# PrP^res in placental tissue following experimental transmission of atypical scrapie in ARR/ARR sheep is not infectious by Tg338 mouse bioassay

Robert B. Piel, III[1,2☯], Valerie R. McElliott[3☯], James B. Stanton[3], Dongyue Zhuang[1], Sally A. Madsen-Bouterse[2], Linda K. Hamburg[1], Robert D. Harrington[1,4], David A. Schneider[1,2]*

1 Animal Disease Research Unit, Agricultural Research Service, United States Department of Agriculture, Pullman, Washington, United States of America, 2 Department of Veterinary Microbiology and Pathology, College of Veterinary Medicine, Washington State University, Pullman, Washington, United States of America, 3 Department of Pathology, College of Veterinary Medicine, University of Georgia, Athens, Georgia, United States of America, 4 Department of Comparative Medicine, University of Washington, Seattle, Washington, United States of America

☯ These authors contributed equally to this work.
* david.schneider1@usda.gov

## Abstract

Nor98-like atypical scrapie is a sporadic disease that affects the central nervous system of sheep and goats that, in contrast to classical scrapie, is not generally regarded as naturally transmissible. However, infectivity has been demonstrated via bioassay not only of brain tissue but also of certain peripheral nerves, lymphoid tissues, and muscle. This study examines placental tissue, a well characterized route of natural transmission for classical scrapie. Further, this study was conducted in sheep homozygous for the classical scrapie resistant ARR genotype and is the first to characterize the transmission of Nor98-like scrapie between homozygous-ARR sheep. Nor98-like scrapie isolated from a United States ARR/ARR sheep was transmitted to four ARR/ARR ewes via intracerebral inoculation of brain homogenate. These ewes were followed and observed to 8 years of age, remained non-clinical but exhibited progression of infection that was consistent with Nor98-like scrapie, including characteristic patterns of PrP^Sc accumulation in the brain and a lack of accumulation in peripheral lymphoid tissues as detected by conventional methods. Immunoblots of placental tissues from the infected ewes revealed accumulation of a distinct conformation of PrP^res, particularly as the animals aged; however, the placenta showed no infectivity when analyzed via ovinized mouse bioassay. Taken together, these results support a low risk for natural transmission of Nor98-like scrapie in ARR/ARR sheep.

## Introduction

Transmissible spongiform encephalopathies (TSEs), or prion diseases, are caused by the misfolding of prion proteins (PrP) from the normal cellular conformation (PrP^C) to the disease

**Data Availability Statement:** All relevant data are within the paper and its Supporting Information files.

**Funding:** DAS, 2090-32000-035-000-D, US Department of Agriculture, Agricultural Research Service, https://www.ars.usda.gov/research/project/?accnNo=431730. The funders had no role in study design, data collection and analysis, decision to publish, or preparation of the manuscript.

**Competing interests:** The authors have declared that no competing interests exist.

conformation (PrP$^{Sc}$) [1]. Accumulation of PrP$^{Sc}$ aggregates, particularly in the brain, results in disease pathogenesis. Prion diseases are progressive, ultimately fatal, and have no known cure [2]. Scrapie, the TSE of sheep and goats, is broadly categorized into two forms, classical and atypical [3–5]. The term atypical scrapie has also been used to refer to prion disease in sheep and goats arising from non-native prions such as bovine spongiform encephalopathy (BSE) and chronic wasting disease (CWD). For the purposes of this manuscript, we use the term atypical scrapie to refer to only the sporadic/spontaneously occurring form of scrapie originally characterized as Nor98 scrapie [6] and known as Nor98-like scrapie for cases in the United States [7].

Classical scrapie was the first agricultural TSE to be described and has been a significant pathogen affecting sheep and goat farming in many countries for centuries. Classical scrapie is highly transmissible, especially at birth when lambs and kids are exposed to prions that have accumulated in the placenta [8–12]. In recent years, significant progress has been made toward the eradication of classical scrapie through the two-fold approach of depopulation of affected animals and selective breeding for genetic resistance [13]. A large body of work demonstrates linkage between certain genetic polymorphisms in the coding sequence of PrP and susceptibility to infection. To date, the combination of polymorphisms: A136, R154, R171 (ARR) has demonstrated the most complete resistance to classical scrapie [14, 15] and is a primary target for breeding programs designed to increase the proportion of resistant genotypes in domestic sheep populations [16].

Historically, atypical scrapie has been regarded as non-transmissible through natural exposure and has therefore been exempted from many of the control measures enacted to combat classical scrapie [16]. However, mounting data suggests that this characterization of atypical scrapie may warrant closer scrutiny. During atypical scrapie, PrP$^{Sc}$ accumulates in the brain and this tissue is infectious to sheep through experimental inoculation by intracranial [17–21] and oral [22] routes. PrP$^{Sc}$ accumulation in peripheral tissues, however, has not been detected by immunoassay [3, 5, 23], which may explain the lack of significant evidence for natural transmission [24]. However, infectivity from peripheral tissues has been documented in the context of transgenic mouse bioassays [19], raising again a concern for potential sources of infective material for natural transmission.

Though selective breeding has proven highly successful in efforts to eradicate classical scrapie, several cases of atypical scrapie have been detected in sheep homozygous for the ARR allele [7, 25, 26]. As this genotype is a foundation of resistance breeding programs, and thus will comprise an increasing proportion of domestic sheep, it is important to characterize fully the potential for atypical scrapie disease progression and transmission in these animals.

In the following study, sheep bearing the ARR/ARR genotype were successfully infected with atypical scrapie isolated from a United States ARR/ARR case [7]. This allowed for a detailed examination of the disease phenotype with respect to progression and PrP$^{Sc}$ distribution both within the brain and in peripheral tissues. This represents the first such study conducted in United States sheep with the classical scrapie-resistant ARR/ARR genotype. The atypical scrapie infected sheep were also bred, and placental tissue was examined to assess the potential for natural transmission of atypical scrapie in ARR sheep under farm conditions.

## Results

### Transmission of atypical scrapie via intracranial inoculation of ARR/ARR sheep

None of the ewes inoculated with the atypical scrapie isolate [7] developed progressive clinical signs of scrapie. One recipient ewe (3913) developed intercurrent disease (abomasal emptying

disorder) and was euthanized at 6 years of age. The other recipient ewes were culled at 8 years of age. In the brain of all four recipient ewes, the regional pattern of PrP$^{Sc}$ accumulation as determined by IHC was characteristic of atypical scrapie (Fig 1A), described in detail below. As expected, accumulation of PrP$^{Sc}$ was not detected by IHC in any of the multiple peripheral lymphoid tissues collected from recipient ewes.

Cerebellum homogenate from each recipient ewe was examined via western blot. All four showed proteinase K (PK)-resistant PrP (PrP$^{res}$) banding patterns matching that of the original inoculum and distinct from the banding pattern associated with classical scrapie (Fig 2A and 2B). Cerebellum homogenates from the original donor and recipient ewes were also analyzed in Tg338 ovinized mouse bioassay. The original case brain homogenate transmitted infection to 7 of 7 passage 1 mice (an eighth mouse was inoculated but could not be tested) with an incubation period ranging from 219 to 284 days. On second passage, infection was again transmitted to all 10 inoculated mice tested by western blot, and yielded a reduced incubation period ranging from 165 to 217 days. Atypical scrapie from the brain of each recipient ewe transmitted infection to all Tg338 mice with an incubation period of 85 to 261 days (Table 1). The PrP$^{res}$ banding pattern in the brain of each infected mouse matched that of the inoculum (Fig 2C and 2D). These results demonstrate successful, experimental transmission and faithful recapitulation of the original atypical scrapie inoculum in ARR/ARR sheep.

## Neuropathologic observations in the brain of recipient ewes

The immunolabeling of PrP$^{Sc}$ and histologic observations were consistent amongst all four recipients except as noted. In general, the described pathologies were least in ewe 3913 (aged 6 years) as compared to the other three ewes (aged 8 years).

G1 (caudal medulla/obex): There was minimal to moderate globular and punctate immunolabeling of the dorsal, lateral and ventral nerve tracts (cuneate fasciculus, spinocerebellar tract, spinal trigeminal tract, pyramids), as well as the medial lemniscus and medial longitudinal fasciculus (Fig 1B). The neuropil of several nuclei and the reticular formation were similarly labeled, but with decreased frequency and a more limited distribution in 3 animals (3913, 3914, 3916). Affected nuclei included external cuneate, spinal trigeminal, spinal vestibular, lateral reticular, hypoglossal, raphe, and rarely, dorsal motor nucleus of the vagus nerve, cuneate, gracilis and inferior olive (dorsal and medial). Intraneuronal immunolabeling was notably absent (Fig 1C). There was a mild to moderate increase in microglia and astrocytes (gliosis) particularly of the dorsal motor nucleus of the vagus nerve, with less involvement of the nucleus of the solitary tract. Rare, shrunken, hypereosinophilic neurons and rounded, hypereosinophilic spheroids were also present in a few of the nuclei (i.e., external cuneate, spinal vestibular, spinal trigeminal, lateral reticular nucleus).

G2 (cerebellar cortex): The cerebellar folia were nearly diffusely affected by segmental to patchy immunostaining. The immunostaining was predominantly coalescing to discrete, finely granular in the molecular layer of all four animals, which often tapered in severity at the interface of the molecular and Purkinje layers (Fig 1D). The immunolabeling was more prominent in the dorsal vermis as compared to the ventral vermis and lateral hemispheres. Three animals (3914, 3916, 3917) had mild punctate immunolabeling of the granular layer (Fig 1D). In all four animals, the neuropil of the deep cerebellar nuclei had rare, widely spaced, punctate and globular immunolabeling. Mild to moderate gliosis (astrocytes and/or microglia) diffusely affected the folia of the vermes and hemispheres. Glial cells often extended to the interface of the molecular and Purkinje cell layers. Additionally, there was mild segmental spongiosis of the Purkinje cell layer with loss of Purkinje cells.

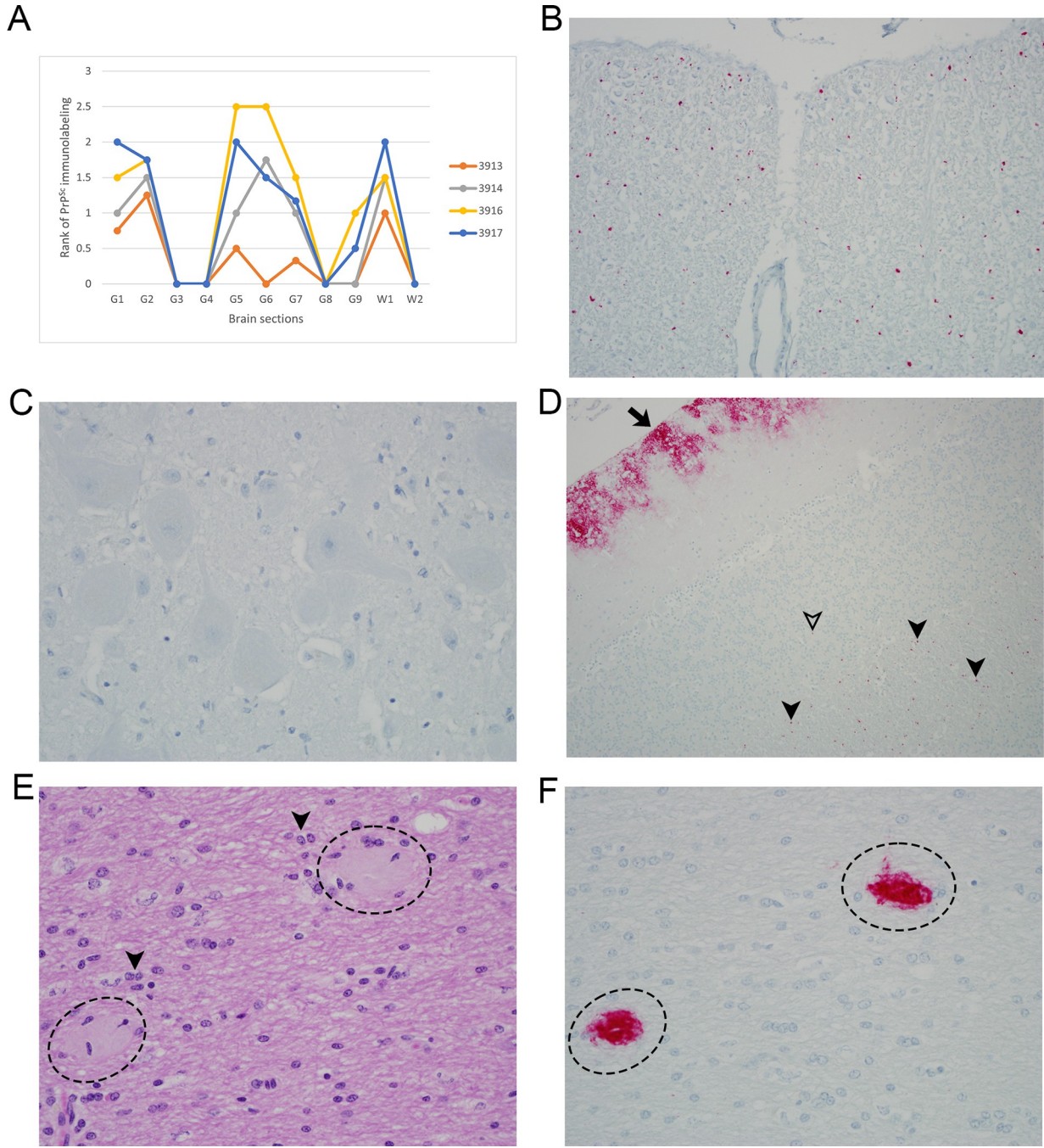

**Fig 1. Accumulations of PrP^Sc in the brain of recipient ewes.** (A) Rank scores of PrP^Sc accumulation for each recipient ewe by brain region: G1 —caudal medulla at the obex, G2—cerebellar cortex, G3—superior colliculus, G4/5—hypothalamus and medial thalamus, G6—hippocampus, G7 —septum, G8—cerebral cortex at level of G4/5, G9—forebrain cortex at level of G7, W1—cerebellar white matter, W2—mesencephalic tegmentum. G1 was plotted as a mean of the rank scores to represent the two brain sections. G6 and G7 were plotted as a mean of the rank scores to include all the gray matter neuroanatomy in the respective section. (B) Punctate to globular extracellular PrP^Sc in spinocerebellar nerve fiber tract. Animal 3917, caudal medulla, IHC. (C) Absent PrP^Sc in the dorsal motor nucleus of the vagus nerve. Animal 3917, caudal medulla, IHC. (D) Patchy but diffuse, coalescing finely granular PrP^Sc in the molecular layer of the cerebellum (solid arrow), and punctate PrP^Sc in the white matter of the folia (solid arrowheads). Rare PrP^Sc in the granular layer (open arrowhead). Animal 3917, cerebellum, IHC. (E) Plaques (dashed ellipses) in the corona radiata with displacement of nerve fibers and mild gliosis (solid arrowheads); H&E. (F) PrP^Sc multicentric aggregates (dashed ellipses) coinciding with the localization of plaques on H&E; IHC. (E, F) Animal 3916, forebrain.

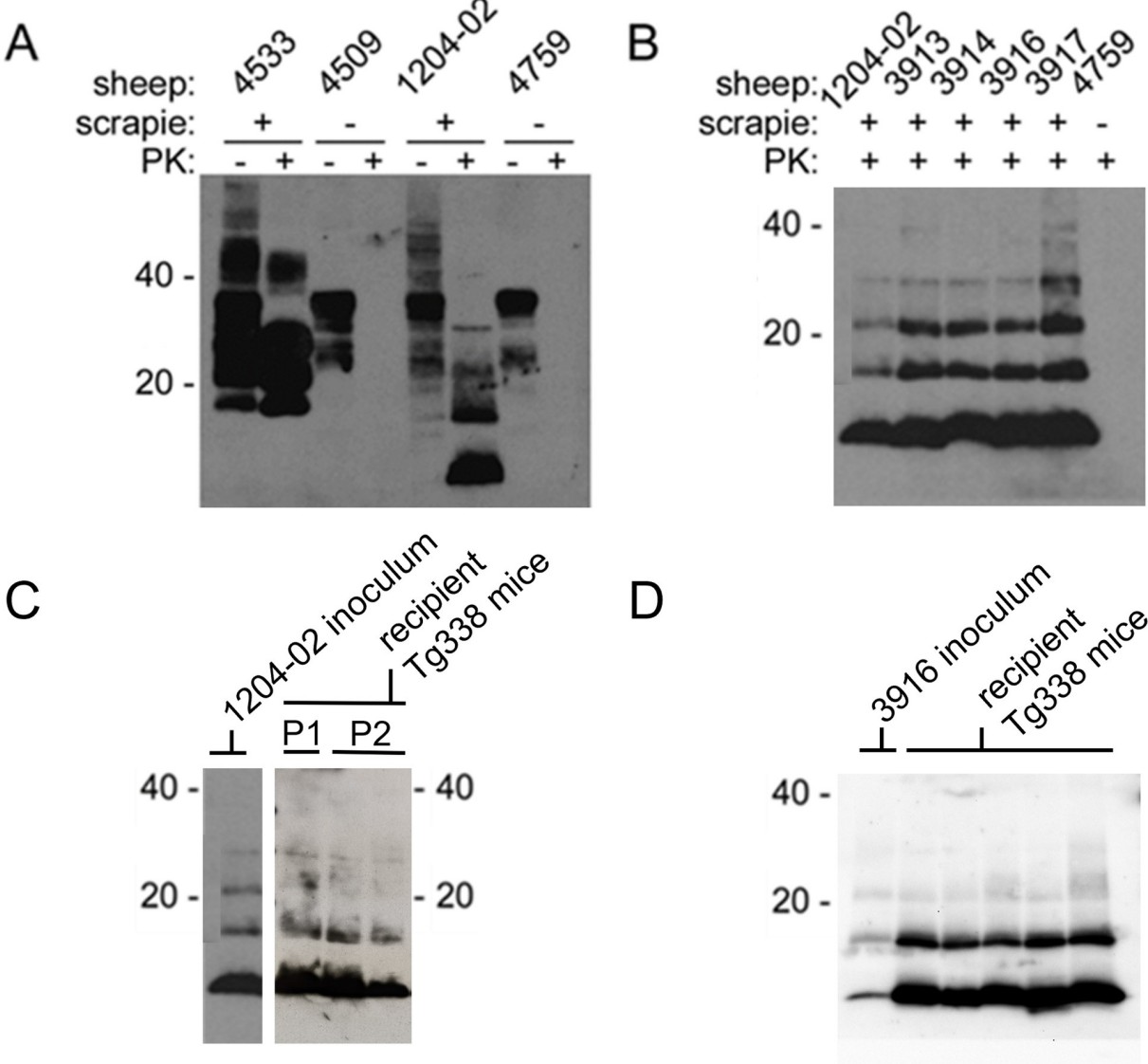

**Fig 2. Western blots demonstrating stable experimental transmission of ARR/ARR atypical scrapie to ARR/ARR sheep and Tg338 ovinized mice.** A) Brain homogenate from the original donor sheep (1204–02) shows an anti-PrP western blot profile indicative of atypical scrapie following treatment with proteinase K (PK). Brain homogenates from a sheep positive for classical scrapie (4533), and scrapie-naïve sheep (4509, 4759) are shown for comparison. B) Western blot of brain homogenates from the original atypical scrapie donor (1204–02), recipient ewes (3913, 3914, 3916, 3917), and a scrapie-naïve sheep (4759). C) Western blot of brain homogenates from Tg338 ovinized mice inoculated with brain homogenate from the original atypical scrapie donor (1204–02). Passage 1 (P1) mice were inoculated with brain homogenate from 1204–02 and Passage 2 (P2) mice were inoculated with brain homogenates from P1 mice. D) Western blot of brain homogenates from Tg338 ovinized mice inoculated with brain homogenate from recipient ewes (representative blot from ewe 3916 shown). Western blots labeled with molecular weight markers of 20, 40 kDa.

G3 (superior colliculus): There was an absence of PrP<sup>Sc</sup> immunolabeling in the caudal midbrain, including the superior colliculus. Additionally, only minimal gliosis, neuronal and axonal changes were observed on H&E in the superior colliculus.

G4/5 (hypothalamus and medial thalamus): The hypothalamic sections lacked immunolabeling. In the thalamus, there were frequent multicentric aggregates and fewer areas of punctate staining in several nuclei (i.e., ventral posteriolateral thalamic, ventral posteriomedial thalamic, ventrolateral, ventral anterior thalamic) with less common immunolabeling in the

**Table 1. Mouse bioassay of brain or placenta cotyledon homogenates (10% w/v).**

| Sheep | Group | Tg338 group[a] | Survival time (dpi)[b] | PrP^res (Positive/Number Inoculated)[c] |
|---|---|---|---|---|
| 1204–02 | Donor | Brain P1 | 219–284 | 7/7 |
| | | Brain P2 | 165–217 | 10/10 |
| 3913 | Recipient | Brain P1 | 126–224 | 10/10 |
| | | Placenta P1 | 497–603 | 0/7 |
| 3914 | Recipient | Brain P1 | 85–226 | 10/11 |
| | | Placenta P1 | 365–601 | 0/7 |
| 3916 | Recipient | Brain P1 | 173–200 | 10/10 |
| | | Placenta P1 | 256–581 | 0/8 |
| 3917 | Recipient | Brain P1 | 183–261 | 10/10 |
| | | Placenta P1 | 338–569 | 0/7 |

Table showing survival times and PrP^res detection in mice intracerebrally inoculated with brain or placental cotyledon homogenates from donor or recipient ewes.

[a] P1 represents first passage inoculation with sheep tissue homogenate. P2 represents second passage inoculation with mouse brain tissue from P1 mice.

[b] Survival time range—days post inoculation (dpi) of mice culled due to the presence of clinical signs (Tg338 inoculated with brain homogenate) or intercurrent issues (Tg338 inoculated with brain or placenta homogenate).

[c] Number of mice with PrP^res by PTA-western blot versus number of mice inoculated.

external medullary lamina, and rare involvement of the internal capsule. This staining consistently resulted in a curvilinear pattern of deposition which extended from the ventromedial thalamus to the dorso-lateral and less frequently dorsal thalamus. Amorphous, lightly eosinophilic, moderately circumscribed, plaques were more frequent on H&E in 3916 and 3917 as compared to 3913 and 3914, which almost always coincided with the localization of PrP^Sc aggregate forms on IHC. On H&E, these aggregates were sometimes surrounded by hypereosinophilic degenerate nerve fibers, few microglia, and occasional lipofuscin-laden microglia or macrophages. There was mild aggregate and punctate immunolabeling of the intralaminar and mediodorsal thalamic nuclei in two animals (3916, 3917), with moderate microgliosis in 3917. Mild microgliosis was in the periventricular and reuniens thalamic nuclei of 3914.

Multifocal to diffuse, periventricular punctate immunolabeling was in the medial amygdaloid region in two animals (3916, 3917). Punctate immunolabeling was also at the angle of the amygdalohippocampus-internal capsule-tail of the caudate nucleus and had a more scattered distribution in the cerebral peduncle of the same animals.

G6 (hippocampus): A predominance of multicentric aggregates, with less frequent finely granular immunolabeling predominated in the dorsal hippocampal commissure at the interface of white and gray matter, and at the medial most extent of the commissure (adjacent to entorhinal cortex) in all animals except for 3913. These aggregates disrupted the ventral, medial and dorsal aspects of the hippocampal commissure. Similar, but less common aggregates were in the dorsal hippocampus between the dentate gyrus and CA1 for the same animals. In 3916, regionally extensive punctate immunolabeling was at the junction of CA3 with the dentate gyrus, and rare punctate foci were superior to CA1. Mild to moderate gliosis (mixture of astrocytes and microglia) partially to circumferentially surrounded these aggregates. There was mild to moderate microgliosis in multiple neuroanatomic locations (i.e., CA3, the junction of the dentate gyrus with CA3, the dentate gyrus, the area between CA1 and the dentate gyrus) with minimal involvement of CA1. The posterior thalamic nuclear group had aggregated and less frequent finely granular immunolabeling. These areas coincided with plaques on H&E. Immunolabeling extended slightly into the midbrain from the posterior thalamic nuclear group in affected animals.

Widely scattered, punctate immunolabeled foci were in the subcallosal bundle and corpus callosum in animals 3916 and 3917, which were infrequent in 3914, and rare in 3913. Focal, globular intraependymal immunolabeling was restricted to one animal (3914). Except for the subcallosal bundle and corpus callosum, additional PrP$^{Sc}$ immunolabeling was absent in the hippocampus and other neuroanatomic regions of the forebrain, in one animal (3913).

G7 (septum): Mild punctate immunolabeling was in the neuropil of the lateral and medial septal nuclei of 3914, 3916 and 3917, with a complete absence of immunolabeling in 3913. In all 4 subjects, multiple neurons in the lateral septal nucleus were enlarged, rounded, and prominent with a vesicular nucleus, and clumped Nissl substance. The distribution of these atypical neurons was diffuse in 3913 and 3914, and multifocal in 3916 and 3917. Additionally, scant to few type II Alzheimer-like astrocytes were often adjacent to these aforementioned neurons. The lateral septal nucleus was infiltrated by mild to moderate gliosis (mixture of astrocytes and microglia). In 3916 and 3917 only, the medial septal nucleus exhibited similar, but milder neuronal and inflammatory changes.

Amongst all animals, there was variation in the type and distribution of extraneuronal immunolabeling in gray matter of the internal capsule. This included finely granular and regionally extensive (3913), punctate and multifocal (3917), multifocal aggregated to finely granular (3914), and widespread punctate to globular (3916). There was immunolabeling in the internal capsule white matter that varied from mild and punctate (3913, 3914), to moderate, punctate and globular (3916, 3917). Immunolabeling was more widespread in 3914, as compared to 3913.

Rare, punctate immunolabeling was in the corpus callosum (3917), and globus pallidus and putamen (3916). Tissue sections of putamen from 3913 and 3914, as well as the globus pallidus from 3914 were absent from examined histologic and immunohistochemical slides.

The lateral olfactory tract had punctate to globular immunolabeling, which extended into the olfactory tubercle in 3913, 3916, and 3917. Although the lateral olfactory tract was absent from the tissue section for 3914, two punctate foci were in the olfactory tubercle.

There was an absence of immunolabeling in the caudate nucleus; however, occasional type II Alzheimer-like astrocytes were in the caudate nucleus of one animal (3917).

G8 (cerebral cortex at level of G4/5): Immunolabeling of the cerebral cortex was confined to the interface between the cortex and the corona radiata in 3914, 3917 and 3916, and was often punctate to globular. Mild, segmental microgliosis was also present at this interface, with slight extension into the cortex in 3916 and 3917.

In the corona radiata, there was variation in the character of PrP$^{Sc}$ immunolabeling amongst some of the animals. The immunolabeling was more widespread in two animals (3916, 3917), comprised of multicentric aggregates largely localized to the gyral tips, and punctate to globular accumulation in the inferior corona radiata. In these same animals, the lateral gyri were most frequently and prominently involved with diminished immunolabeling of the medial/middle gyri. The locale of aggregates in the lateral gyri often coincided with plaques on H&E in 3916 (Fig 1E and 1F), but these plaques were less overt on H&E in 3917. There was also mild microgliosis in the inferior aspect of the corona radiata of some gyri. Animal 3916 also had mild, scattered perivascular edema.

In animal 3914, the immunolabeling in the corona radiata was punctate to globular in the gyral tips with rare punctate accumulation in the inferior corona radiata. Lastly, animal 3913 had only rare punctate immunolabeling that was confined to the periventricular white matter, and a focal intraventricular aggregate adhered to the apical surface of a single ependymal cell.

G9 (forebrain cortex at level of G7): Immunolabeling of the forebrain cortex was rare, but when present, was often punctate occurring at the interface of the cerebrocortex and corona radiata in two animals (3916, 3917) with slight extension into the cortex. No significant histopathologic lesions were in the cortex of any animal.

PrP$^{Sc}$ immunolabeling of the corona radiata in two animals (3916, 3917) was most pronounced in gyral tips, tapering in severity at the inferior base (adjacent to lateral ventricle). The pattern varied from mainly multicentric aggregates to less common globular and punctate immunolabeling. There were plaques on H&E in the corona radiata of gyral tips, that occasionally coincided with aggregates on IHC. Variable numbers of astrocytes and microglia surrounded these aggregates. Rare, punctate immunolabeling was widely scattered in the corona radiata of 3914, and there was an absence of immunolabeling in 3913. Additionally, there was moderate, widespread, punctate periventricular immunolabeling (3916, 3917), and focal, globular immunolabeling of intraventricular debris (3913).

W1 (cerebellar white matter): The white matter of the folia had globular and punctate immunolabeling, with extension into but tapering of severity at the level of the ventral vermis (Fig 1D). Occasional globular, periventricular immunolabeling was also present. No significant histopathologic lesions were in the cerebellar white matter in any of the four animals.

W2 (mesencephalic tegmentum): There was an absence of immunolabeling, as well as significant histopathologic lesions in the mesencephalic tegmentum. However, the superior cerebellar peduncle possessed rare, globular immunolabeling in two animals (3913, 3914).

## Accumulation of PrP$^{res}$ in placenta of recipient ewes

Ewes infected with atypical scrapie were bred each year and up to 6 cotyledons were collected from each placenta and examined for PrP$^{Sc}$ accumulation. Each cotyledon was divided in two with one half being sectioned and evaluated via PrP$^{Sc}$ immunohistochemistry and the other half tested by western blotting. None of the placentas tested showed PrP$^{Sc}$ accumulation in cotyledons by IHC (Fig 3A). However, western blot did show accumulation of protease resistant PrP (PrP$^{res}$) in cotyledons of some placentas (Fig 3B). The PrP$^{res}$ detected in these placentas displayed variable banding patterns, but within the same molecular weight range as that of brain homogenates from the recipient ewes and original donor (Figs 2B and 3B). PrP$^{res}$ was detected in 22 of 93 cotyledons (and 9 of 20 placentas) tested. Notably, the proportion of cotyledons exhibiting PrP$^{res}$ accumulation increased with the age of the recipient ewes (Fig 3D and 3E).

PrP$^{res}$ accumulation in placental cotyledons from each recipient ewe (Fig 3B) was tested for infectivity via ovinized mouse bioassay. While brain tissue from mice inoculated with brain homogenate from the original donor and recipient ewes did demonstrate infectivity and showed western blot banding patterns characteristic of atypical scrapie (Fig 2C and 2D), brains from mice inoculated with cotyledon homogenates did not show evidence of prion accumulation (Fig 3C).

## Discussion

In this study, we document the experimental transmission of Nor98-like atypical scrapie, where both the donor and recipient sheep are homozygous for the ARR allele. This genotype is a major target for selective breeding programs to combat classical scrapie, and thus it is imperative to understand the way in which atypical scrapie strains/ecotypes behave and progress in these animals.

Immunohistochemical analysis of the brains of recipient ewes confirmed the experimental transmission of atypical scrapie to all four recipient ewes. In contrast to sheep infected with classical scrapie, these atypical scrapie-infected ewes had a predominance of PrP$^{Sc}$ accumulation in the lateral nerve tracts of the medulla, molecular layer of the cerebellum, corona radiata, thalamic nuclei, and hippocampal commissure, and absent or only mild accumulation of PrP$^{Sc}$ in the dorsal motor nucleus of the vagus nerve, granular layer of the cerebellum, superior colliculus, and hypothalamus.

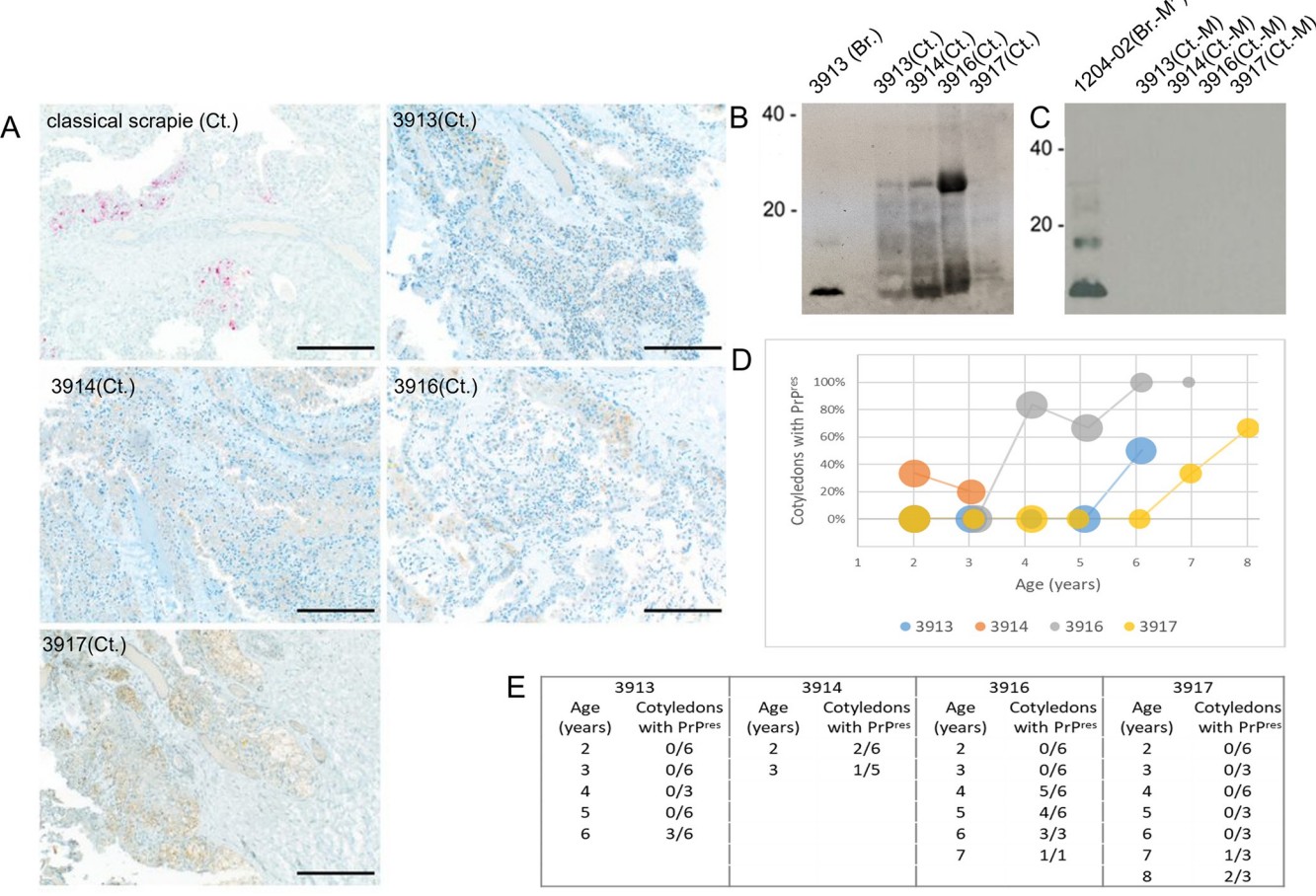

**Fig 3. Accumulation of non-infective PrP^res in the placentas of ARR/ARR ewes infected by atypical scrapie from a US ARR/ARR sheep.** A) PrP^Sc was not detected in cotyledons (Ct.) from infected ewes (3913, 3914, 3916, 3917) by immunohistochemical staining; note red staining in cotyledon from a sheep with classical scrapie shown for comparison. B) Anti-PrP western blot showing accumulation of PrP^res in cotyledons from infected ewes. Brain homogenate (Br.) from ewe 3913 shown for comparison. C) Anti-PrP western blot showing lack of PrP^res in brain homogenates from Tg338 mice inoculated with homogenates of cotyledons containing PrP^res (Ct.–M); inocula were prepared from the same cotyledons shown in (B). Brain homogenate of passage 2 mouse (Br.–M*) from original atypical scrapie donor shown for comparison. Western blots labeled with molecular weight markers of 20, 40 kDa. D,E) The proportion of cotyledons containing PrP^res, detected via western blot, increased with the ages of the inoculated ewes.

Overall, the PrP^Sc localization and patterns of PrP^Sc deposition are similar to those previously described for atypical scrapie [7, 21, 22, 27–29], but a few differences were noted. The hippocampal commissure and the dorsal hippocampus had aggregated and finely granular immunolabeling compared to previous reports of finely granular to nondisclosed type of immunolabeling in the hippocampus (the hippocampal commissure was not specifically mentioned in previous studies) [30–32]. Additionally, finely granular immunolabeling was diffusely observed in up to three laminar layers of the neocortex in 3 separate brain lobe sections in a previous study [27], whereas our data show punctate to globular immunolabeling was largely restricted to the interface of the gray and white matter with rare involvement of the cortex in only 2 brain lobe sections. In other studies [7, 22, 27–29], finely granular to punctate to nondisclosed type of immunolabeling was in the caudate nucleus, whereas there was a complete absence of immunolabeling in the caudate nucleus in the current study. Also, punctate immunolabeling has been noted in the olfactory tract [22], but in the samples examined here, punctate immunolabeling was also in the olfactory tubercle (olfactory tubercle not specifically

mentioned in previous studies). Additionally, we found that cerebellar cortical PrP$^{Sc}$ immuno-labeling was largely confined to the molecular layer, but in previous reports [7, 27–29], there was also involvement of the granular layer. Additionally, only low amounts of punctate and granular immunolabeling were identified in the thalamus, midbrain, and cerebral cortex in a previous study [21], whereas in the current study, large, multicentric aggregates were frequent in the medial thalamus, the caudal midbrain lacked immunolabeling, and cerebral cortical labeling was localized to the interface of the corona radiata with the cerebral cortex.

Some of these variations from previous descriptions could relate to differences in the stage of infection or route of exposure. Additionally, genotypic polymorphisms in *PRNP* are known to influence susceptibility to atypical scrapie infection [3, 33]. It is also known that phenotypic expression can vary even within the same genotype, suggesting other molecular and cellular factors may also contribute [27]. Previous studies [7, 21, 22, 27–29] encompass a range of conditions including preclinical and clinical disease in animals either naturally infected or experimentally inoculated with atypical scrapie. These studies also cover a variety of *PRNP* genotypes, including combinations of AHQ, ARQ, AFRQ, ARH, and ARR alleles. The present study is the first to characterize PrP$^{Sc}$ distribution following atypical scrapie transmission between ARR/ARR animals. The results from the present study suggest, that progression to clinical disease and the more advanced pathologies described are not likely to be present if natural transmission of atypical scrapie occurs between ARR/ARR sheep during a typical commercial lifespan.

Atypical scrapie is commonly regarded as non-transmissible under natural conditions with PrP$^{Sc}$ considered to be confined to the central nervous system [7, 22, 27]. As such, there exists no regulatory framework for the control of atypical scrapie in domestic ruminants. Concerningly, previous studies have reported infectivity in ovinized mice inoculated with peripheral tissues of sheep naturally infected with atypical scrapie, despite these tissues being IHC negative for PrP$^{Sc}$ [19]. Peripheral tissues demonstrated to carry infectivity included peripheral nerves, several lymphoid tissues, and the external ocular muscle [19]. Though these tissues represent areas of concern in the context of rendering and subsequent food chain re-entry [34], they are relatively unlikely to be a direct source of prion shedding to the environment. But infected lymphoid tissues could be an indirect source for shedding if the prions demonstrated in lymphoid tissues can escape this tissue in the lymph. Carried in the lymph, prions would ultimately flow into the general blood circulation from where they potentially could escape the body in secretions and excretions or as shed with the placenta at birth.

Placental tissue is known to be a major route of horizontal transmission and environmental contamination for classical scrapie under natural conditions [8–12]. To investigate the possible risks of placenta-mediated transmission, cotyledons from recipient ewes were initially examined by IHC and western blot. While no evidence of PrP$^{Sc}$ accumulation was seen via IHC, western blotting did reveal the presence of protease resistant PrP (PrP$^{res}$) in cotyledons. We do not classify this material more specifically than PrP$^{res}$ due to its distinct western blot banding pattern compared to that observed in the brain homogenates of infected sheep and mice (Figs 2B–2D and 3B). It is possible that the differential banding patterns are due to tissue context differences between brain and placenta. Interestingly, the proportion of cotyledons containing PrP$^{res}$ increased as the ewes aged. This pattern of increasing PrP$^{res}$ accumulation over subsequent pregnancies is reminiscent of that observed in cases of classical scrapie [9, 11]. However, when this material was used to inoculate Tg338 mice, no infectivity was observed. This suggests the PrP$^{res}$ observed in the cotyledons is phenotypically distinct from that found in brain samples from the same animals.

The inability of classical scrapie-resistant genotypes to offer protection from atypical scrapie [7, 25, 26], coupled with the spontaneous/sporadic nature and worldwide distribution of

atypical scrapie, have raised concerns over the potential for phenotype shifts and related transmission. One study documented the emergence of a strain of classical scrapie in an AHQ/AHQ sheep inoculated with atypical scrapie from an ARR/ARR sheep [35]. Other studies in pigs [36] and bovinized mice [37] have shown phenotype shifts to BSE-like prion strains upon experimental transmission of atypical scrapie. Over the course of this study, no instances of phenotype shift were observed in any of the sheep or ovinized mice inoculated with atypical scrapie, including second passage Tg338 mice (Fig 2B–2D). While the limitations of this experiment cannot preclude the risk of phenotype/strain shifts, it contributes to the overall dataset in this area of study.

In summary, this experiment represents the first experimental transmission of Nor98-like atypical scrapie between sheep homozygous for the ARR allele. No evidence of strain/phenotype shifts were observed and though accumulation of PrP$^{res}$ was observed in placentas from inoculated ewes, this material was non-infectious in mouse bioassay, consistent with a low risk for natural transmission associated with this tissue.

## Material and methods

### Animal use and care assurance

All care and use of the sheep and transgenic mice in this study was approved by the Institutional Animal Care and Use Committee at Washington State University (protocols 1649, 3815, 4175, 4267, 4924, 6096, 6561). Sheep were euthanized by intravenous injection of pentobarbital-based euthanasia solution. Mice were euthanized by inhalation of carbon dioxide followed by cervical dislocation.

### Original Nor98-like scrapie case material

The material used for inoculation was a 10% w/v homogenate of the cerebellum from a non-clinical, eight-year-old Suffolk ewe (donor ewe, 1204–02) that had been in quarantine for five years in a mixed group of sheep acquired by USDA from flocks exposed to natural cases of classical scrapie. This ewe is identified as case two in the original report of the first six cases of Nor98-like scrapie confirmed in the U.S. [7]. The donor ewe was homozygous at all *PRNP* codons, including methionine at codon 112, alanine at codon 136, lysine at codon 141, and arginine at codons 154 and 171; i.e., homozygous MALRR.

### Experimentally infected breeding (recipient) ewes

Five weaned ewe lambs (four Columbia, one Rambouillet) homozygous for the *PRNP* haplotype MALRR were obtained from the US Sheep Experiment Station, Dubois, Idaho. At 1 year of age, the ewes were needle inoculated with 1 mL of original case homogenate by intracerebral route. Four recipient ewes recovered without complications; one Columbia ewe was not recovered due to seizure activity during inoculation. The recovered ewes were monitored daily by farm staff for general health and signs of scrapie, and monthly by a veterinarian (DAS) for signs of scrapie and body condition scoring [38]. Breeding was by natural cover each fall. Rams (Suffolk, Polypay, mixed) were acquired for breeding purposes from sources naïve to classical scrapie and *PRNP* genotyped as homozygous MALRR or, when this genotype was not available in a timely manner, heterozygous for QR at codon 171 (MALRR/MALRQ). Hereafter, these genotypes are referenced simply as ARR/ARR or ARR/ARQ representing codons 136, 154 and 171.

Stage 3 labor (expulsion of the placenta) was attended, when possible, to facilitate acquisition of intact placenta. Placentas were refrigerated up to 24 hours prior to processing.

Cotyledons were collected from each fetal-maternal unit and either fixed in 10% neutral-buffered formalin or frozen at -20˚C.

Postmortem tissues collected for this study included the whole brain, retropharyngeal lymph node (RPLN), palatine tonsils, nictitating membrane of the third eyelid, ileum, ileocecal junction, ileocecal and other regional mesenteric lymph nodes, rectoanal mucosa, spleen, caudal mediastinal lymph node, supramammary lymph node, popliteal lymph node, prefemoral lymph node, and prescapular lymph node. Each brain was divided into two halves, one side fixed in neutral-buffered 10% formalin, the other frozen at -80˚C. Tissues frozen at -20˚C included RPLN, tonsil, ileocecal lymph node, and spleen.

## Immunohistochemistry (IHC)

Scrapie IHC was performed as previously described for cotyledons [11] and other tissues [39], with minor modification. In brief, fixed tissues were treated with formic acid in cassettes for 1 hour before paraffin embedding, or for 5 minutes as paraffin sections mounted on glass slides. Sections were cut at 5 μm and used for scrapie IHC. A serial section from each brain region was also used for routine histopathology (hematoxylin and eosin stained, H&E). An automated slide staining system was used to perform deparaffinization, antigen retrieval, and immunolabeling using mouse monoclonal antibodies F99/97.6.1 [40] and F89/160.1.5 [41] (5 μg/mL, each), a universal secondary antibody, fast red chromogen kit, and hematoxylin counterstaining. Control samples were matched tissue from both scrapie infected and uninfected animals used at the start and end of each run.

## Evaluations of recipient brains

Multiple sections of brain were assessed for PrP$^{Sc}$ localization and accumulation using previously established protocols [42, 43], with minor modifications. Nine areas of gray matter and two areas of white matter were evaluated, that includes: G1 (caudal medulla at the obex); G2 (cerebellar cortex); G3 (superior colliculus); G4/5 (hypothalamus and medial thalamus); G6 (hippocampus); G7 (septum); G8 (cerebral cortex at level of G4/5); G9 (forebrain cortex at level of G7); W1 (cerebellar white matter); and W2 (mesencephalic tegmentum). An additional section of more caudal medulla at the level of the central canal was also evaluated. Brains from two sheep with classical scrapie were used for comparison of histopathologic lesions and immunohistochemical staining patterns.

All slides were evaluated independently by two board-certified veterinary pathologists (VRM, JBS), as a single blind-trial. A consensus was derived for any interpretations in which there was an initial discrepancy. In H&E sections, the presence of frequent neuropil vacuolation was noted; however, the lack of age-matched controls precluded the ability to definitively score or interpret the validity of this lesion. Additional lesions (i.e., plaque-like aggregates, gliosis, neuronal necrosis, and neuronal vacuolation) were also evaluated and recorded.

PrP$^{Sc}$ immunolabeling was evaluated in all brain sections, and the character of the immunolabeling was recorded utilizing the nomenclature previously implemented [27]. PrP$^{Sc}$ accumulation was graded on a subjective scale of 0–3 (0 = absence of immunolabeling; 1 = mild immunolabeling; 2 = moderate immunolabeling; 3 = intense immunolabeling) based on the localization and severity for each brain section. For any given level of sectioning, IHC was performed on all experimental animals in the same run.

## Mouse bioassay

Brain tissue from all sheep and cotyledons from shed placentas were tested for infectivity via bioassay in Tg338 mice.

Inocula for mouse bioassay were prepared as previously described [44]. In summary, homogenates of cerebellum were prepared at 10% w/v in sterile phosphate-buffered saline and homogenates of placental cotyledons were prepared at 10% w/v in sterile 0.9% saline. Homogenates were filtered through sterile gauze and gentamycin added to a final concentration of 100 μg/ml. Homogenates were tested for bacterial contamination and, if needed, heat sterilized as described previously [45].

Mice were inoculated intracerebrally with 20 μl homogenate each, as described previously [44]. Mice were monitored for clinical signs of TSE disease including weight loss, lethargy, extreme limb weakness, ataxia, tremors, bilaterally closed eyes, and kyphosis. All mice in this study were culled upon observation of clinical signs of TSEs, intercurrent disease, or age-related health concerns prior to the predetermined study endpoint of 700 days post inoculation. Mouse brains were collected upon euthanasia and stored at -20˚C.

Second passage (P2) mouse bioassay was performed for the original donor ewe inoculum (1204–02), wherein brain homogenate prepared from the first passage (P1) mouse bioassay was used to inoculate P2 mice.

## Western blotting

Cerebellum and mouse brain homogenates were prepared at 10% in phosphate-buffered saline. Homogenates of placental cotyledons were prepared at 10% or 20% w/v in 0.9% saline.

Prior to analysis by western blot, PrP$^{Sc}$ in tissue homogenates was concentrated by precipitation with sodium-phosphotungstic acid (NaPTA) [45–47]. Homogenates were treated with 4.0 mg/mL collagenase at 37˚C for 120 min, then mixed 1:1 with 4% w/v sarkosyl and further incubated at 37˚C for 15 min. Next, samples were treated with 100 μg/ml DNase at 37˚C for 45 min. Samples were then centrifuged at 1400 x $g$ for 8 min to remove particulate matter. NaPTA stock solution (4% w/v NaPTA in 170 mM MgCl$_2$, pH 7.4) was added to resulting supernatants to a final concentration of 0.3% NaPTA and samples incubated 37˚C for 75 min. Next, samples were treated with 20–50 μg/ml proteinase K at 37˚C for 25–60 min. PrP$^{Sc}$ was pelleted by centrifugation at 19000 x $g$ for 20 min and the pellet subsequently resuspended in 10 mM Tris-HCl (pH 7.5), 0.5% NP-40, 0.5% sodium deoxycholate. For some samples containing larger amounts of PrP$^{Sc}$, an additional centrifugation step was added prior to proteinase K treatment.

Following NaPTA precipitation, samples were mixed with NuPAGE LDS Sample Buffer and Reducing Agent (Invitrogen) to 1X strength according to manufacturer's instructions, boiled for 10 min, run by electrophoresis through 12% Bis-Tris protein gels (Invitrogen), and transferred to PVDF membranes via semi-dry transfer. Membranes were blocked in Blocker Casein (Pierce/Thermo Scientific). Anti-PrP primary antibodies used include 3.5 μg/ml F99/97.6.1 and mAb L42 (RIDA-Biopharm) diluted 1:700 in Blocker Casein. Anti-Mouse IgG (H+L) Antibody, F(ab')2 Fragment, Human Serum Adsorbed and Peroxidase-Labeled (KPL) was diluted 1:6000 in Blocker Casein and used as the secondary antibody. Signal was detected via chemiluminescence using either the Amersham ECL (GE Healthcare) kit on film (Carestream Health BioMax™ Light Film; Fisher Scientific) or the ChemiGlow West Chemiluminescence Substrate Kit (Protein Simple) and a FluorChem R imager (Protein Simple).

## Supporting information

**S1 Raw images.**
(PDF)

## Acknowledgments

The authors wish to thank the following people for providing excellent technical support to this project: Tom Truscott, Lori Fuller, Laetisha O'Rourke, Jian Zhang. The authors greatly appreciate the consistently excellent animal care provided by numerous staff members over the extended years of this project. Special appreciation is given to Dr. Katherine O'Rourke who participated in the initial design and management of this study. Mention of trade names or commercial products in this article is solely for the purpose of providing specific information and does not imply recommendation or endorsement by the US Department of Agriculture.

## Author Contributions

**Conceptualization:** James B. Stanton, Sally A. Madsen-Bouterse, Robert D. Harrington, David A. Schneider.

**Data curation:** Robert B. Piel, III, Valerie R. McElliott, Dongyue Zhuang, Sally A. Madsen-Bouterse, Linda K. Hamburg.

**Formal analysis:** Robert B. Piel, III, Valerie R. McElliott, James B. Stanton, Dongyue Zhuang, Sally A. Madsen-Bouterse, David A. Schneider.

**Funding acquisition:** David A. Schneider.

**Investigation:** Robert B. Piel, III, Valerie R. McElliott, James B. Stanton, Dongyue Zhuang, Sally A. Madsen-Bouterse, Linda K. Hamburg, Robert D. Harrington, David A. Schneider.

**Methodology:** Robert B. Piel, III, Valerie R. McElliott, James B. Stanton, Dongyue Zhuang, Sally A. Madsen-Bouterse, Linda K. Hamburg, Robert D. Harrington, David A. Schneider.

**Project administration:** David A. Schneider.

**Supervision:** James B. Stanton, Sally A. Madsen-Bouterse, David A. Schneider.

**Validation:** Robert B. Piel, III, Valerie R. McElliott, Sally A. Madsen-Bouterse, David A. Schneider.

**Visualization:** Robert B. Piel, III, Valerie R. McElliott, Sally A. Madsen-Bouterse, Linda K. Hamburg, David A. Schneider.

**Writing – original draft:** Robert B. Piel, III, Valerie R. McElliott, James B. Stanton, David A. Schneider.

**Writing – review & editing:** Robert B. Piel, III, Valerie R. McElliott, James B. Stanton, Dongyue Zhuang, Sally A. Madsen-Bouterse, Linda K. Hamburg, Robert D. Harrington, David A. Schneider.

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
