## [Decision Letter · Decision Letter 0]

17 Nov 2021

PONE-D-21-31945PrPres in placental tissue following experimental transmission of atypical scrapie in ARR/ARR sheep is not infectious by Tg338 mouse bioassay.PLOS ONE

Dear Dr. Schneider,

Thank you for submitting your manuscript to PLOS ONE. After careful consideration, we feel that it has merit but does not fully meet PLOS ONE’s publication criteria as it currently stands. Therefore, we invite you to submit a revised version of the manuscript that addresses the points raised during the review process.

Please  address all of the comments provided by the reviewers in a revised manuscript and in a point-by-point response to the reviewers' comments.

We look forward to receiving your revised manuscript.

Kind regards,

Mark Zabel

Academic Editor

PLOS ONE

Journal Requirements:

2. PLOS ONE has specific criteria regarding the reporting of animal research (https://journals.plos.org/plosone/s/submission-guidelines#loc-animal-research). Specifically, these guidelines require that details regarding care, monitoring, and method of sacrifice are clearly stated. As part of your revision, please complete and submit a copy of the ARRIVE Guidelines checklist, a document that aims to improve experimental reporting and reproducibility of animal studies for purposes of post-publication data analysis and reproducibility: https://arriveguidelines.org/sites/arrive/files/Author%20Checklist%20-%20Full.pdf. Please include your completed checklist as a Supporting Information file. Note that if your paper is accepted for publication, this checklist will be published as part of your article.

[The authors wish to thank the following people for providing excellent technical support to this project: Tom Truscott, Lori Fuller, Laetisha O’Rourke, Jian Zhang. The authors greatly appreciate the consistently excellent animal care provided by numerous staff members over the extended years of this project. Special appreciation is given to Dr. Katherine O’Rourke who participated in the initial design and management of this study. This work was funded by the USDA-Agricultural Research Service under CRIS 2090-32000-035-000-D. Mention of trade names or commercial products in this article is solely for the purpose of providing specific information and does not imply recommendation or endorsement by the US Department of Agriculture.]

 [DAS, 2090-32000-035-000-D, US Department of Agriculture, Agricultural Research Service, https://www.ars.usda.gov/research/project/?accnNo=431730. The funders had no role in study design, data collection and analysis, decision to publish, or preparation of the manuscript.]

Reviewers' comments:

Reviewer's Responses to Questions

**Comments to the Author**

1. Is the manuscript technically sound, and do the data support the conclusions?

Reviewer #1: Yes

Reviewer #2: Yes

2. Has the statistical analysis been performed appropriately and rigorously? 

Reviewer #1: N/A

Reviewer #2: N/A

3. Have the authors made all data underlying the findings in their manuscript fully available?

Reviewer #1: Yes

Reviewer #2: Yes

4. Is the manuscript presented in an intelligible fashion and written in standard English?

Reviewer #1: Yes

Reviewer #2: Yes

5. Review Comments to the Author

Reviewer #1: PrPres in placental tissue following experimental transmission of atypical scrapie in ARR/ARR sheep is not infectious by Tg338 mouse bioassay.

Corresponding Author: Dr. D.A. Schneider

First Author: R.B. Piel III

The authors demonstrate experimental intracranial (IC) transmission of Nor98-like scrapie to ARR/ARR scrapie resistant sheep by demonstration of PK resistant PrP (PrPres) in brain tissue assayed by immunohistochemistry (IHC). PrPres accumulation within the brain is similar to that previously reported for atypical scrapie strains, with a few differences as described by the authors. Upon ovinized mouse bioassay they confirm the presence of the infectious scrapie agent within brain tissue harvested from the IC-inoculated sheep. The authors further examine placental tissue harvested from this experimental sheep transmission study for PrPres and infectivity. They report presence of PrPres within placental tissues, yet lack of infectivity as analyzed by ovinized mouse bioassay.

The manuscript is well written and organized.

Minor Comments:

Line 91-99 and Lines 431-435: Are the sheep infection studies/tissues used for this manuscript harvested from sheep inoculated in the original publication (Reference 7; Loiacono et. al.)? If so, please mention in Materials and Methods.

Line 121 and line 127: Maintain consistency in reporting Tg or tg.

Lines 392-394: Do atypical scrapie prions circulate in the periphery via blood transport? As the authors point out there is presence of infectivity in lymphoid tissue harvested from atypical scrapie cases. This reviewer agrees that ‘they are relatively unlikely to be a source of prion shedding’, yet, as infectivity is present in lymphoid tissue, it may spillover into the peripheral blood supply to organs associated with prion shedding in saliva, urine, feces, etc. This could result in the shedding of PK resistant prions in bodily fluids/excretions to the environment. Please comment within the discussion that this may be the case.

Lines 401-404 and 408-409: The difference may also be due to the route of inoculation, IC, vs a more natural route of inoculation/infection. Please indicate within the discussion the impact the route of inoculation may have on your results.

Reviewer #2: Hello,

This is an exceptionally well-written and well-presented manuscript. My only suggestions are minor and consist of requests to add symbolic annotations (arrows, arrowheads, etc...) to figures 1D and 1E to point out the PrPsc accumulations in 2 regions (1D) and plaques and glial cells (1E).

6. PLOS authors have the option to publish the peer review history of their article (what does this mean?). If published, this will include your full peer review and any attached files.

Reviewer #1: No

Reviewer #2: No

---

## [Author Response · Author response to Decision Letter 0]

3 Dec 2021

See "Response to Reviewers.docx" file attached.

---

## [Decision Letter · Decision Letter 1]

5 Jan 2022

PrPres in placental tissue following experimental transmission of atypical scrapie in ARR/ARR sheep is not infectious by Tg338 mouse bioassay.

PONE-D-21-31945R1

Dear Dr. Schneider,

We’re pleased to inform you that your manuscript has been judged scientifically suitable for publication and will be formally accepted for publication once it meets all outstanding technical requirements.

Kind regards,

Mark Zabel

Academic Editor

PLOS ONE

Additional Editor Comments (optional):

Reviewers' comments:

Reviewer's Responses to Questions

**Comments to the Author**

1. If the authors have adequately addressed your comments raised in a previous round of review and you feel that this manuscript is now acceptable for publication, you may indicate that here to bypass the “Comments to the Author” section, enter your conflict of interest statement in the “Confidential to Editor” section, and submit your "Accept" recommendation.

Reviewer #2: All comments have been addressed

2. Is the manuscript technically sound, and do the data support the conclusions?

Reviewer #2: Yes

3. Has the statistical analysis been performed appropriately and rigorously? 

Reviewer #2: Yes

4. Have the authors made all data underlying the findings in their manuscript fully available?

Reviewer #2: Yes

5. Is the manuscript presented in an intelligible fashion and written in standard English?

Reviewer #2: Yes

6. Review Comments to the Author

Reviewer #2: (No Response)

7. PLOS authors have the option to publish the peer review history of their article (what does this mean?). If published, this will include your full peer review and any attached files.

Reviewer #2: No

---

## [Editor Report · Acceptance letter]

10 Jan 2022

PONE-D-21-31945R1 

PrP^res^ in placental tissue following experimental transmission of atypical scrapie in ARR/ARR sheep is not infectious by Tg338 mouse bioassay 

Dear Dr. Schneider:

I'm pleased to inform you that your manuscript has been deemed suitable for publication in PLOS ONE. Congratulations! Your manuscript is now with our production department. 

Kind regards, 

on behalf of

Dr. Mark Zabel 

Academic Editor

PLOS ONE